# Tracking Visual Programming Language-Based Learning Progress for Computational Thinking Education

**Ting-Ting Wu** [1] , **Chia-Ju Lin** [2] , **Shih-Cheng Wang** [2] **and Yueh-Min Huang** [2,*]

1. Graduate School of Technological and Vocational Education, National Yunlin University of Science and Technology, Yunlin 64002, Taiwan
2. Department of Engineering Science, National Cheng Kung University, Tainan City 70101, Taiwan
* Correspondence: huang@mail.ncku.edu.tw

**Abstract:** Maker education that incorporates computational thinking streamlines learning and helps familiarize learners with recent advances in science and technology. Computational thinking (CT) is a vital core capability that anyone can learn. CT can be learned through programming, in particular, via visual programming languages. The conclusions of most studies were based on quantitative or system-based results, whereas we automatically assessed CT learning progress using the Scratch visual programming language as a CT teaching tool and an integrated learning tracking system. The study shows that Scratch helped teachers to diagnose students' individual weaknesses and provide timely intervention. Our results demonstrate that learners could complete tasks and solve problems using the core CT steps. After accomplishing numerous tasks, learners became familiar with the core CT concepts. The study also shows that despite increased learning anxiety when solving problems, all learners were confident and interested in learning, and completed each task step by step.

**Keywords:** computational thinking; visual programming language; Scratch; learning tracking system

## 1. Introduction

With the global rise of the maker movement, governments across the world have begun to focus on the impact of maker activities on learners. The philosophy of maker education is learning by making; this indicates a transformation from the conventional dissemination of knowledge to learning relevant concepts by doing. That is, maker education allows students to implicitly acquire knowledge while completing maker projects. This method of learning has a positive and effective impact on learners. Compared with conventional teaching methods, students think about and apply their knowledge, during which they proactively identify and address problems instead of acquiring knowledge passively.

Scientific, manufacturing, and other technological advances have reduced the costs of maker equipment. Also, as the maker movement has developed, embedded development systems have matured, and sensors have become more affordable and diverse, helping learners develop their creativity. Information technology (IT) has resulted in new development and applications for maker education, and it is now cheaper and easier to integrate maker education into on-site education than it once was. These developments all significantly benefit learners. Countries all over the world have developed maker education, and the U.S. Department of Education is cooperating with Exploratorium in offering maker courses to high-poverty and low-performing regions [1]. In Europe, the Fabrication Laboratory (FabLabs), the EU-initiated MakerSpace, and organizations such as Maker Faire Rome and Startup Europe provide maker spaces for learners to tap into their creativity. In Taiwan, the movement has been nurtured by the Workforce Development Agency, which has established maker bases and factories. In addition to hardware developments, the maker concept has been brought into the classroom. Maker education requires not only

hardware investments but also integrated software, as well as teachers and teaching materials, helping learners to gain a better understanding of the potential value of the maker movement. Furthermore, students can be trained to acquire existing knowledge from tasks.

The literature shows that maker education integrates well with emerging science and technology, such as STEAM education [2,3], virtual reality (VR) [4], and computational thinking (CT) [5]. This gives learners access to emerging science and technology in a better and faster manner, reflecting today's rapidly changing culture. Additionally, the literature indicates that maker education assists students in developing their creativity, collaborative skills, and problem-solving abilities [6] and improves their engagement in classes [7], all of which have long-term, vertical impacts on learners rather than short-term impacts [8].

CT has been a popular research topic in recent years [9]. In 2006, Professor Wing of CMU indicated that CT is a basic skill needed in daily life and that CT is a key element for elementary education. She re-defined CT and showed that it is just as important as the "3 Rs" (reading, writing, and arithmetic) and that every child should be encouraged to hone their analytical skills using CT [10]. CT is a thinking process in which people use basic concepts and logical methods from computer science to identify and seek solutions step by step [10,11]. Accordingly, learning CT helps us tackle problems more effectively, understand root causes, and address more sophisticated problems [12,13]. In addition, the increasing importance of CT has motivated countries throughout the world to implement CT training policies [14,15].

CT is generally learned through programming [16,17]. Though current programming languages closely resemble natural languages, abstract concepts that are implicit in text-based programming languages are difficult for beginners to learn [18]. In contrast to such text-based programming languages, visual programming languages (VPLs) present language structures via visual blocks of different colors and shapes. This enables beginners to design programs by manipulating blocks, thus significantly lowering the threshold of programming [19]. Relevant studies demonstrate that VPL is an effective learning method for CT, which explains the increasing use of VPL in CT education using systems such as Scratch [20] and Blocky [21]. Scratch facilitates user-defined block-based design to design programs using VPLs and also connects with IoT devices. Therefore, Scratch is the most popular learning instrument for CT [22]. Despite the fact that many studies have attested to the effectiveness of VPL for learning CT [22,23], most determined it by quantitative methods [24,25] involving CT tests or scales [25–27]. Some assessed student programming projects through operating systems [25,28]. However, such methods failed to comprehensively analyze the programming and learning processes and thus did not investigate students' operations during visual programming (VP). It is also important to effectively and automatically assess the learning effectiveness of CT [29] and determine whether students understand CT, particularly in problem-solving.

We thus tapped the Scratch VPL as a programming language tool to teach CT and developed a learning and tracking system for CT education. This system facilitated real-time tracking of programming projects and tasks, allowing teachers to grasp the learning pace of every student as well as the various project results. The system logged the writing procedures and paths of students during programming assignments to help teachers diagnose students' learning weaknesses. Timely intervention and assistance then alleviated students' anxiety and boosted learning motivation and confidence. We posed the following research questions:

(1) After participating in the course of Scratch programming, is there any difference in the frequency of using computational thinking skills?
(2) Does participation in the Scratch programming course affect learning motivation, learning anxiety, and learning confidence?

## 2. Research Method

### 2.1. Participants

To equip subjects with basic computer and programming skills, we recruited fifth- and sixth-grade students from elementary schools as participants. Twenty-eight participants (16 boys and 12 girls) voluntarily participated in this course. With the help of CT concepts, the participants completed tasks and challenges by using the Scratch-based learning platform while the system recorded and visualized their progress in real time. The teacher had teaching experience in using Scratch and participated in the overall course design and planning.

### 2.2. Experimental Design

Figure 1 shows the experimental procedures of this study. The Scratch programming course was taught in two 40-min sections each week for six weeks. In the first week, the teacher explained the learning procedures and programs to give students a preliminary understanding of the Scratch tasks. The teacher explained the core CT concepts and steps so that learners would grasp course structure, after which the teacher demonstrated and explained the learning system's functions and procedures to familiarize students with the operating environment and prepare them for follow-up courses. In addition, students completed questionnaires and scales about learning motivation, learning confidence, and learning anxiety to record their own feelings before the activities. The experimental activities commenced in the second week. The teacher explained the course content and samples and showed how to complete the problem-solving tasks step by step via the four core CT steps (decomposition, pattern recognition, abstraction, and algorithms). An example description is shown in Table 1. After the teacher explained the tasks, the learners logged into the system to complete the tasks: five basic tasks and five advanced tasks, including maze and math problems, as shown in Table 2. The question types all reflected questions designed by Chien [30] and were modified to suit the activities so that students could use Scratch to complete the tasks and solve the problems according to the core CT steps. The tasks and events lasted for four weeks, and post-tests and interviews were administered in the last week, during which students completed questionnaires and scales again. The learners' viewpoints and thoughts concerning the teaching activities, procedures, and content were then solicited via interviews.

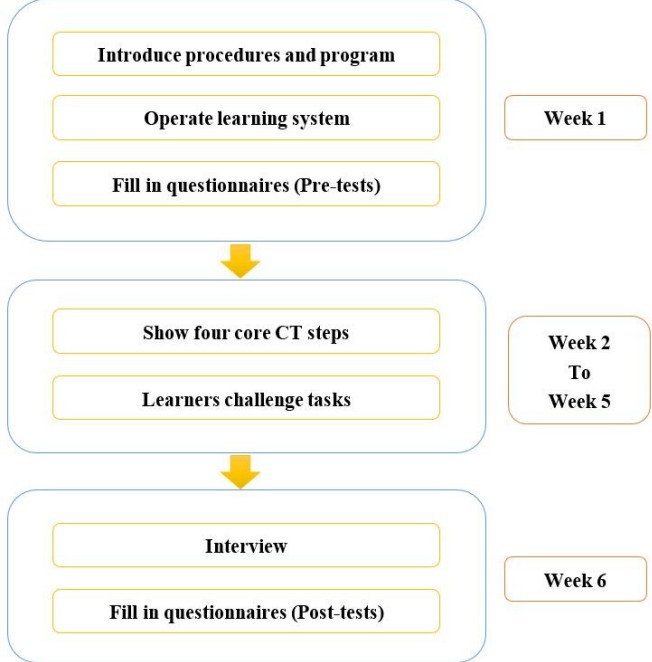

**Figure 1.** Experimental procedure of this study.

**Table 1.** Complete the problem-solving tasks via the four core CT steps.

| Core CT Step | Step Statement | System Interface |
|---|---|---|
| Decomposition | 1. Polar bears cannot hit icebergs<br>2. A polar bear must touch a fish to successfully eat it<br>3. Polar bears can only go where the blue water is | |
| Pattern recognition | 1. Two steps to the right<br>2. Two steps down | |
| Abstraction | 1. Two steps to the right: building blocks to the right<br>2. Two steps down: building down blocks | |
| Algorithms | Use program blocks to control the polar bear, have the polar bear move in the order of the blocks | |

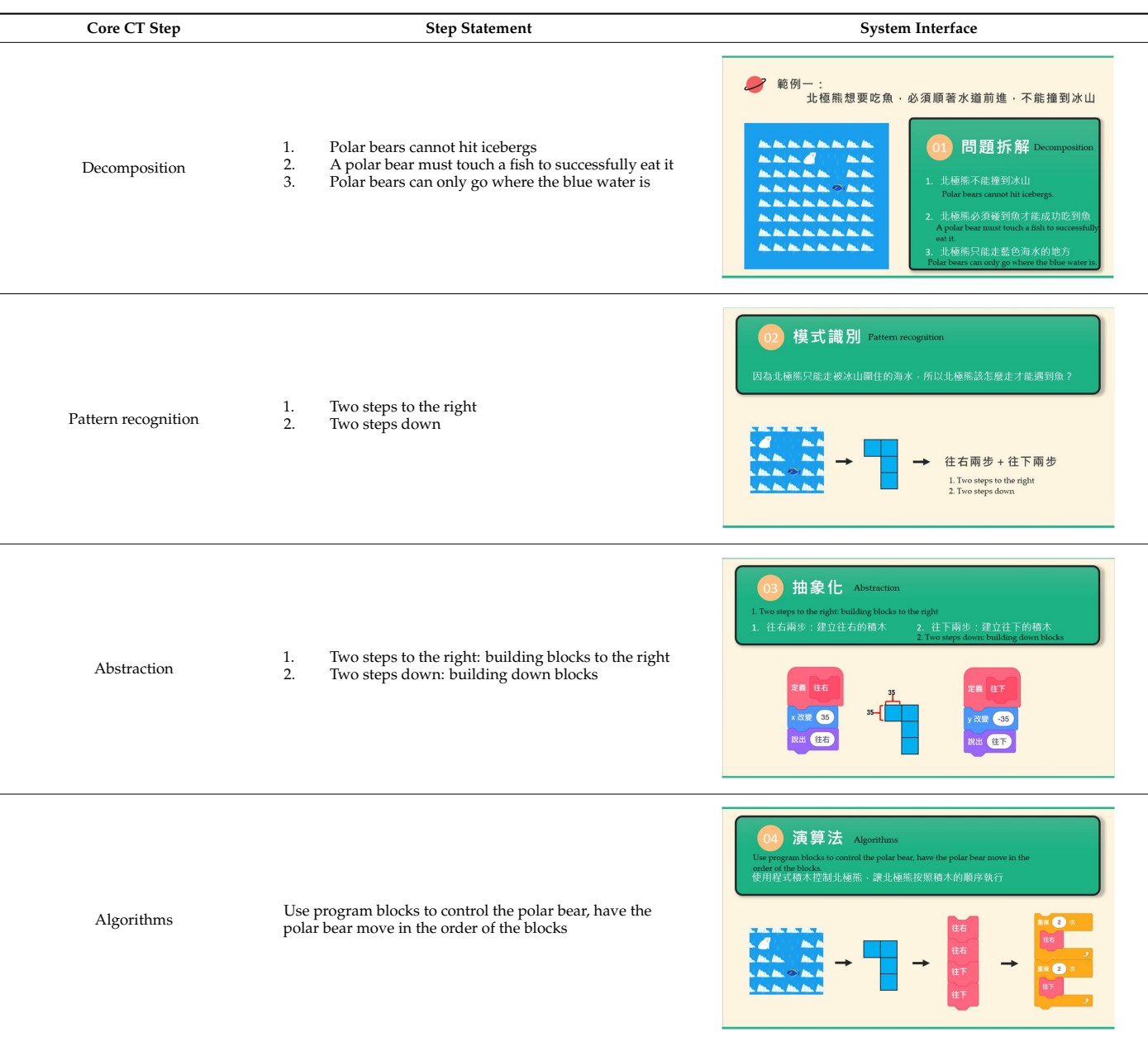

**Table 2.** Five basic tasks and five advanced tasks of this study.

| Level | Task | Statement | System Interface |
|---|---|---|---|
| Basic task 1 | Maze problem: polar bear eating fish | Complete tasks using blocks with defined topics | |

**Table 2.** *Cont.*

| Level | Task | Statement | System Interface |
|---|---|---|---|
| Basic task 2 | Maze problem: polar bear eating fish | Use topic blocks and repeat blocks to complete tasks |  |
| Basic task 3 | Maze problem: polar bear eating fish | Define blocks and repeat blocks to complete tasks |  |
| Basic task 4 | Draw a rectangle | Repeat blocks to draw rectangles |  |
| Basic task 5 | Draw three squares | Define blocks and repeat blocks to draw three squares |  |
| Advanced task 1 | Maze problem: polar bear eating fish | Complete tasks with a limited number of blocks |  |
| Advanced task 2 | Maze problem: polar bear eating fish | Use conditional judgment blocks to complete tasks |  |
| Advanced task 3 | Maze problem: polar bear eating fish | Use specified blocks and conditional judgment blocks to complete tasks |  |

| Level | Task | Statement | System Interface |
|---|---|---|---|
| Advanced task 4 | Draw five triangles | Define blocks and repeat blocks to draw five triangles | |
| Advanced task 5 | Automated obstacle-avoiding vehicle | Define building blocks and use conditions to judge building blocks to complete tasks | |

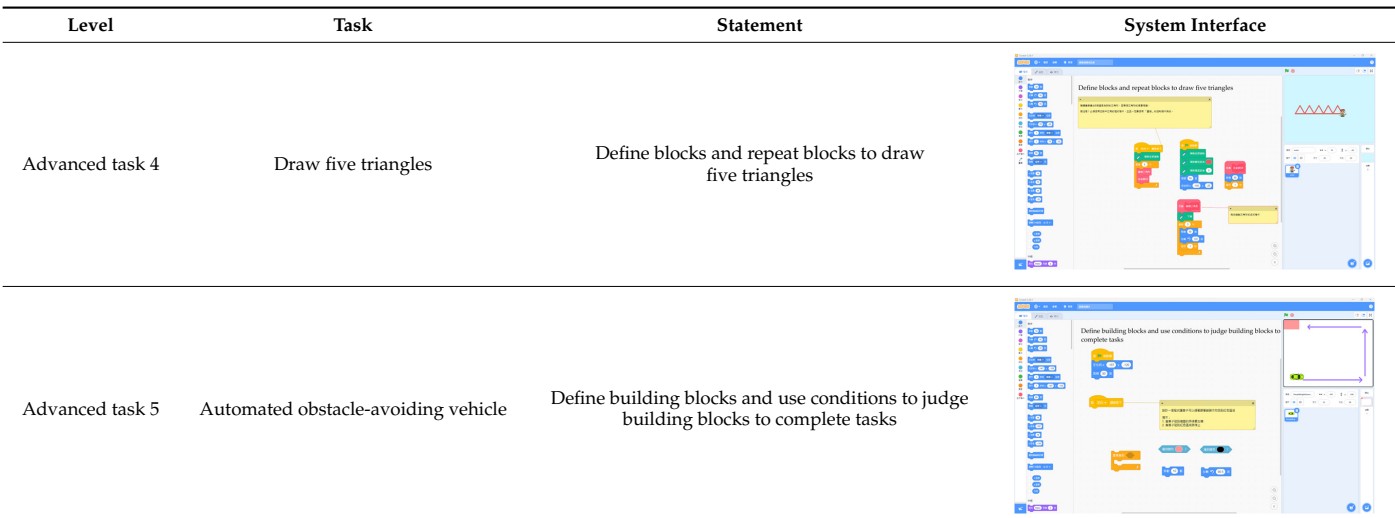

### 2.3. Learning Platform

We customized the interfaces and functions of the VP platform in this study in line with the research design and created cross-platform learning environments for VP using website features, as shown in Figure 2. The Scratch interface was divided into six major parts: (1) block programming categories; (2) blocks; (3) canvas; (4) staging area; (5) roles; and (6) tools. From those parts, students reflected on and selected the desired block programming category and then dragged the programs to the canvas to combine the blocks. After constructing the program, they clicked the green flag to run it or the red button to suspend it in the staging area to preview the logical results of the program.

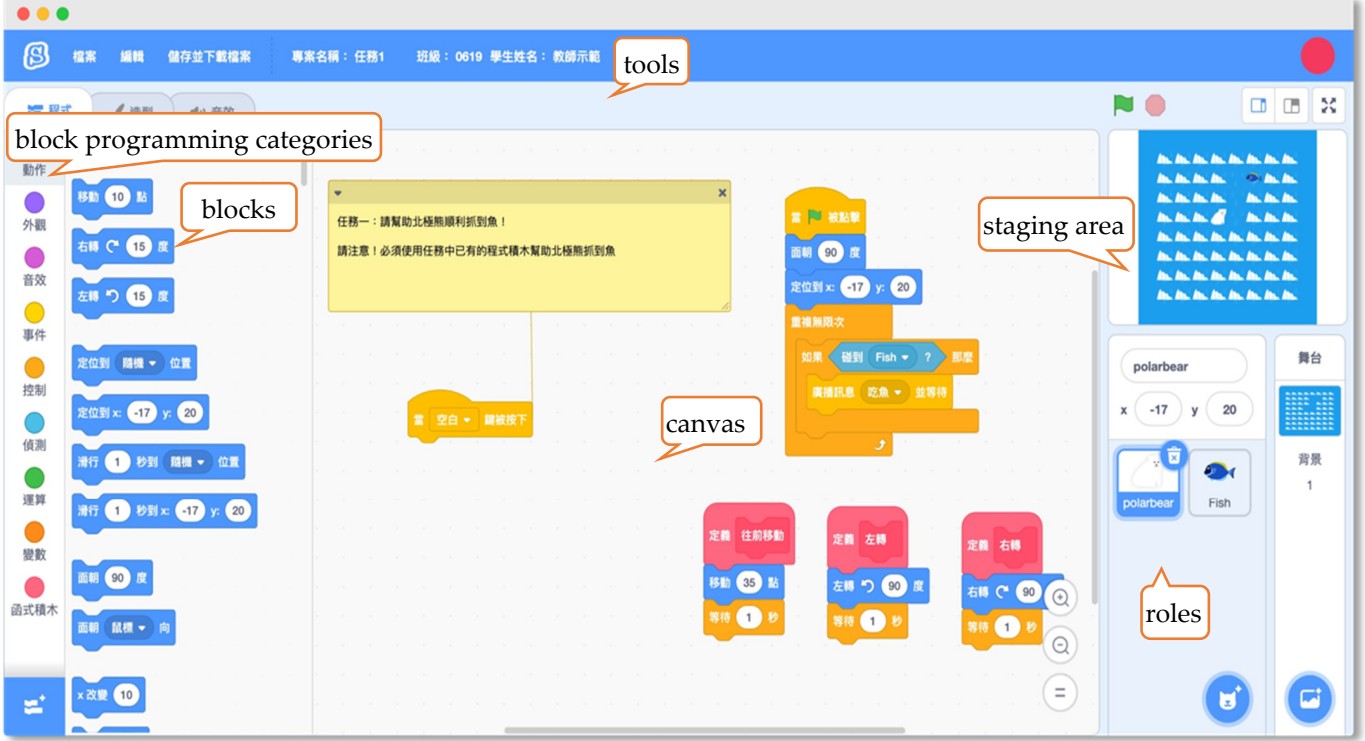

**Figure 2.** Interfaces and functions of the VP platform in this study.

We assembled the integrated learning and tracking system on the Scratch VP platform. The system recorded the time-stamped operations of students. Once these records were transformed in the backend system, students' behaviors were extracted from the logs, and the files were stored in the cloud servers. We used coding and analysis tools to better understand students' behavior when completing the programming tasks and to determine the role that the platform played in the learning process. The behavior logs not only recorded the students' actions during programming tasks but also grasped the creation of them via the coding and analysis instrument. This helped teachers to provide timely assistance to students or adjustments to the course content.

*2.4. Assessment Tools*

1. Frequency of CT Skill Use (FCT)

In this study, we modified the Frequency of Using CT Skills Questionnaire (FCTQ) by Yin, Hadad [31] to evaluate the results of our experiment. We analyzed the reliability of the questionnaire, yielding a Cronbach's $\alpha$ of 0.939. The scale had nine items in total to understand how frequently the learners use CT to think and solve problems in the implementation of task-challenge activities after the teacher explained the four core steps of CT.

2. Learning anxiety

To understand student anxiety during the tasks and Scratch challenges, we used a modified Computer Anxiety Scale Venkatesh [32] ($\alpha = 0.887$). Of the nine items on the scale, five were inverse items to measure learner anxiety about the courses and tasks.

3. Learning motivations

We used the Motivated Strategies for Learning Questionnaire (MSLQ) Pintrich [33] ($\alpha = 0.981$) which was divided into an intrinsic motivation part and an extrinsic motivation part, each with four questions. This allowed us to better understand the impact of proper intervention and assistance on learners' motivation.

4. Learning confidence

We used the Computer Attitudes and Confidence Questionnaire [34] ($\alpha = 0.975$) which included 8 questions to understand the learning self-confidence of the participants before and after the experiments.

5. Interviews

During the experiment of the previous week, we randomly interviewed four learners to solicit their impressions and thoughts on the activity design, course content, and system interface, as face-to-face interviews yielded better understanding of the students' standpoints. The interviews were recorded and videotaped, and the interview results were utilized to assist when quantitative materials were insufficient.

## 3. Research Results

*3.1. FCTQ*

The *t*-test (as shown in Table 3), showed that the FCTQ questions were all significant ($p < 0.05$) at a confidence level of 95%. FCTQ used a five-point Likert scale. The larger the value means the more frequent the use.

This questionnaire revealed that learners gained an in-depth understanding of the CT steps and concepts and completed the tasks and solved the problems using the four core steps. Teachers took advantage of the system's behavior tracking to offer real-time instruction and assistance to familiarize students with CT concepts and help learners apply them to solve the problems. As the learners completed various tasks, they became more familiar with CT steps and concepts, increasing their CT usage frequency and even transferring what they had learned to solve problems in daily life.

**Table 3.** t-test results of FCTQ questions.

| Items | N | Mean | SD | df | t | p |
|-------|---|------|-----|-----|--------|------|
| CT_F1 | 28 | 4.50 | 0.51 | 27 | 46.77 * | 0.00 |
| CT_F2 | 28 | 4.36 | 0.49 | 27 | 47.25 * | 0.00 |
| CT_F3 | 28 | 4.43 | 0.50 | 27 | 46.50 * | 0.00 |
| CT_F4 | 28 | 4.46 | 0.51 | 27 | 46.51 * | 0.00 |
| CT_F5 | 28 | 4.39 | 0.50 | 27 | 46.74 * | 0.00 |
| CT_F6 | 28 | 4.46 | 0.51 | 27 | 46.51 * | 0.00 |
| CT_F7 | 28 | 4.46 | 0.51 | 27 | 46.51 * | 0.00 |
| CT_F8 | 28 | 4.39 | 0.50 | 27 | 46.74 * | 0.00 |
| CT_F9 | 28 | 4.43 | 0.50 | 27 | 46.50 * | 0.00 |

* $p < 0.05$.

### 3.2. Learning Anxiety

As shown in Table 4, a paired samples *t*-test for the learning anxiety questionnaire revealed significant differences in the pre- and post-tests of items ($p < 0.05$) at a confidence level of 95%. The anxiety questionnaire used a five-point Likert scale. Larger values mean more anxiety. The average value indicates that the post-test results were greater than the pre-test results, which shows that the learning anxiety of the learners during the task challenges was greater than that before starting the tasks. This suggests that the task content and problem-solving methods increased the students' anxiety. Nevertheless, the post-test average ranged from 2.5 to 2.9, which is acceptable. A suitable amount of learning anxiety enhanced learning effectiveness [35]. This suggests that the teaching content, tasks, teaching procedures, and system presentation were suitable for students at this age.

**Table 4.** Paired samples' t-test results for learning anxiety analysis.

|  | Mean | | SD | | | |
|-------|------|------|------|------|---------|------|
| Items | Pre | Post | Pre | Post | t | p |
| A1 | 1.54 | 2.96 | 0.51 | 0.84 | −10.95 * | 0.00 |
| A2 | 1.46 | 2.82 | 0.51 | 0.61 | −12.85 * | 0.00 |
| A3 | 1.57 | 2.82 | 0.50 | 0.67 | −11.30 * | 0.00 |
| A4 | 1.54 | 2.82 | 0.51 | 0.55 | −14.79 * | 0.00 |
| A5 | 1.54 | 2.93 | 0.51 | 0.72 | −11.72 * | 0.00 |
| A6 | 1.57 | 2.89 | 0.50 | 0.69 | −12.76 * | 0.00 |
| A7 | 1.54 | 2.89 | 0.51 | 0.61 | −10.23 * | 0.00 |
| A8 | 1.46 | 2.64 | 0.51 | 0.62 | −11.38 * | 0.00 |
| A9 | 1.46 | 2.53 | 0.51 | 0.64 | −7.91 * | 0.00 |

* $p < 0.05$.

### 3.3. Learning Motivation

Motivation is a complicated mental process composed of intrinsic interest, attitudes and desires, the selection of people for experience and targets, and a driving force that influences behavioral attitudes and results in changes in those attitudes [36]. As shown in Table 5, a paired samples *t*-test of the results revealed significant differences between the eight items of learning motivations ($p < 0.05$) at a confidence level of 95%. The learning motivation questionnaire used a five-point Likert scale. Larger values mean more motivation. The post-test results were greater than those of the pre-test (Mean). Thus, the lively and friendly interfaces of systems and task challenges increased the students' learning motivation during the activities. Moreover, the interfaces not only improved extrinsic motivation but also familiarized the students with the system and helped them complete their tasks. The tasks proceeded from easy to difficult. Meanwhile, being able to solve each problem along with the peer pressure stimulated learners' intrinsic motivation, promoting learners to use CT for problem-solving. Repeated practice and thinking helped learners gain proficiency in the CT steps.

**Table 5.** Paired samples' t-test results of learning motivation analysis.

| | Mean | | SD | | | |
|---|---|---|---|---|---|---|
| **Items** | **Pre** | **Post** | **Pre** | **Post** | **t** | **p** |
| A1 | 3.21 | 4.39 | 0.83 | 0.69 | −11.38 * | 0.00 |
| A2 | 3.29 | 4.36 | 0.85 | 0.62 | −9.38 * | 0.00 |
| A3 | 3.25 | 4.50 | 0.89 | 0.51 | −12.76 * | 0.00 |
| A4 | 3.29 | 4.50 | 0.85 | 0.51 | −12.89 * | 0.00 |
| A5 | 3.32 | 4.46 | 0.82 | 0.58 | −13.49 * | 0.00 |
| A6 | 3.36 | 4.57 | 0.78 | 0.50 | −15.38 * | 0.00 |
| A7 | 3.21 | 4.43 | 0.83 | 0.50 | −12.89 * | 0.00 |
| A8 | 3.32 | 4.50 | 0.82 | 0.51 | −13.11 * | 0.00 |

\* $p < 0.05$.

### 3.4. Learning Confidence

Confidence, a measure of self-assessment adjusted to the environment [37,38], is closely related to self-efficacy, the belief in the mastery of our own capabilities and competence. We conducted analyses using paired samples' t-tests, yielding the results in Table 6. The learning confidence questionnaire used a five-point Likert scale. Larger values mean more confidence. There were significant differences in all items ($p < 0.05$) at a confidence level of 95%, and the results of post-tests were better than those of pre-tests (Mean). We thus observed that detailed CT steps and programs helped learners solve problems efficiently while bolstering their confidence. When students got stuck, teachers were able to directly address their weaknesses with real-time assistance, which helped lower the students' frustration level and boost their confidence. This helped students to focus on solving the problem at hand.

**Table 6.** Paired samples' t-test results of learning confidence analysis.

| | Mean | | SD | | | |
|---|---|---|---|---|---|---|
| **Items** | **Pre** | **Post** | **Pre** | **Post** | **t** | **p** |
| A1 | 3.18 | 4.32 | 0.82 | 0.67 | −10.23 * | 0.00 |
| A2 | 3.29 | 4.36 | 0.81 | 0.62 | −10.51 * | 0.00 |
| A3 | 3.21 | 4.43 | 0.83 | 0.50 | −11.31 * | 0.00 |
| A4 | 3.25 | 4.46 | 0.84 | 0.51 | −12.89 * | 0.00 |
| A5 | 3.29 | 4.46 | 0.81 | 0.58 | −15.99 * | 0.00 |
| A6 | 3.36 | 4.57 | 0.78 | 0.50 | −15.38 * | 0.00 |
| A7 | 3.21 | 4.39 | 0.79 | 0.57 | −13.11 * | 0.00 |
| A8 | 3.32 | 4.50 | 0.82 | 0.51 | −13.11 * | 0.00 |

\* $p < 0.05$.

### 3.5. Findings from Interviews

We coded and summarized the interviews with the four students, finding that the learners were satisfied with the system design: the system was easy to use, and the content was at an appropriate level for the students. However, for the tasks, although participants possessed basic computer and programming skills, it took time to become familiar with the four core CT steps for problem-solving. In particular, when students observed their peers completing the tasks, they felt stressed, which manifested as learning anxiety.

Some students mentioned that when they got stuck, the teacher provided timely instruction and assistance and was aware of the steps and procedures that the students had used earlier. This assistance relieved the students and alleviated their anxiety. Furthermore, the tasks helped them to be more familiar with CT: solving the various problems increased their confidence and their intrinsic and extrinsic motivation and helped them to further devote themselves to problem-solving.

## 4. Discussion

We developed a CT learning platform and established a learning and tracking system for Scratch. We used the behavior records to better understand the learners' VPL programming approaches for various tasks. The analysis results helped teachers understand the students' programming progress and offer timely support when necessary. CT problem-solving not only improved the logical skills and systemic thinking of learners but also enhanced their problem-solving skills, which were equally valuable for problems encountered in daily life.

The FCT analysis showed that most learners understood the four core CT steps and solved the problems based on CT concepts. Nonetheless, when they attempted to solve the problems for the first time, they were not able to master the CT steps. Different attempts to solve the problems and tasks only increased their learning anxiety. Since the learners' progress was tracked and recorded in the backend database, the teacher was able to know what difficulties each student faced by consulting their logs, and was thus able to render timely assistance and offer helpful instructions. Such assistance relieved anxiety generated during tasks, improving the students' motivation and confidence.

In addition, learners mastered the four core CT steps and concepts as they completed the tasks. Learning anxiety did occur during the process, but a certain amount of anxiety is necessary to enhance learning efficiency [35]. Moreover, using the concept of computational thinking could improve the logic and systematisms of learners; learners could also apply computational thinking in the process of solving problems in the future and even effectively achieve the effect of learning transfer. Solving the various problems familiarized students with the CT steps, boosted their confidence, and taught them to keep trying in the face of difficulty. Timely assistance and explanations from the teacher also helped students maintain their motivation and gave them the confidence to complete the tasks.

The results of this study are consistent with the drive theory [39], in which the intrinsic drive motivates intrinsic physiological needs, resulting in behavior. Motivation is indispensable for learning. When the needs of individuals are not met, the internal drive is stimulated, leading to reactions. Needs must be satisfied to achieve the desired result. Accordingly, teachers consulted the system's learning logs and analyses to understand the programming progress of each student and offered instruction and assistance to meet their needs. Repeated success in solving problems enhanced learners' confidence and motivation, and the desire to complete more tasks stimulated internal drive, motivating students to accomplish the tasks. The inverted U-shape theory [40] also supports these results: there is a U-shape curvilinear relationship between learning performance and anxiety. As anxiety rises, performance will gradually improve, and when anxiety rises to a certain level, the best performance will be produced. In the tasks in this experiment, learner anxiety was maintained at an acceptable range, promoting learners' confidence and interest in the system activities and helping them to complete the tasks.

## 5. Conclusions and Recommendations for Future Work

In this study, we developed a CT learning platform and established a learning and tracking system for Scratch. The system tracked and recorded programming activities during the course so that teachers could better understand each student's progress and difficulties. Using this learning and analysis system, teachers could offer timely assistance and relevant explanations to students at different levels. The analysis results show that all participants were able to complete the tasks and solve the problems according to the four core CT steps. When the students faced difficulties, the teacher offered instructions and assistance to reduce their anxiety. Additionally, various tasks served to familiarize the students with CT, and problem-solving boosted their confidence and motivation.

From interviews, we learned that the course was too fast-paced. As a result, though participants had basic computer and programming skills, they still needed time to familiarize themselves with the four core CT steps for problem-solving. In addition, we learned that the proposed CT learning platform has many potential applications. Specifically, the

learning and tracking system is well-suited for higher education. The CT-based problem-solving, pattern recognition, abstraction, and algorithm design can be incorporated into course designs to train students' logical thinking and problem-solving capabilities.

**Author Contributions:** Conceptualization, T.-T.W. and S.-C.W.; methodology, T.-T.W. and S.-C.W.; validation, C.-J.L.; formal analysis, T.-T.W. and S.-C.W.; investigation, T.-T.W. and S.-C.W.; writing—original draft preparation, T.-T.W. and C.-J.L.; Writing—review and editing, T.-T.W. and Y.-M.H.; supervision, Y.-M.H. All authors have read and agreed to the published version of the manuscript.

**Funding:** This research was partially supported by the Ministry of Science and Technology, Taiwan under Grant No. MOST 110-2511-H-224-003-MY3 and MOST 111-2628-H-224-001-MY3.

**Institutional Review Board Statement:** Not applicable.

**Informed Consent Statement:** Not applicable.

**Data Availability Statement:** The data presented in this study are available upon reasonable request from the corresponding author, except Experimental Research Participation Consent Form and Declaration of Parental Consent.

**Conflicts of Interest:** The authors declare no conflict of interest.

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
