# Peer review of "Tracking Visual Programming Language-Based Learning Progress for Computational Thinking Education"

_sustainability, doi:10.3390/su15031983_

Round 1

Reviewer 1 Report

1. I suggest to change the title, as in the current version it is really obscure. Typically, in English we don't use more than 3 adjectives describing a noun.

2.  The titles of the sections should not be at the bottom of the previous page, when the content starts on the next one.

3. What is "p" in the statement "p < .05"? All introduced variables must be defined and explained.

4. Also, I suggest to give fractions with "0" in the front. So instead of .05 it should be 0.05. It is much clearer.

5. Regarding section 5 ("Conclusion and Recommendations...") I think that the first paragraph should be moved to the previous section, as it does not convey any general thoughts, but details referring to the experiments. Also, I think that the last concluding paragraph should be rephrased to sound better in English, and to be sure that the authors wrote what they meant. I don't think the word "moreover" is appropriate when the authors jump from "the course too compact" to "potential in applications". 

Author Response

  1. I suggest to change the title, as in the current version it is really obscure. Typically, in English we don't use more than 3 adjectives describing a noun.

Thank you for the suggestion. We have changed the title and hired an English editing company to go through the manuscript.

  1. The titles of the sections should not be at the bottom of the previous page, when the content starts on the next one.

We have adjusted this, as suggested by the reviewer.

  1. What is "p" in the statement "p < .05"? All introduced variables must be defined and explained.

Associated with every t-value is a p-value that represents the probability that the results from the sample data have occurred by chance. P-values range from 0% to 100% and are usually written as a decimal (for example, a p-value of 5% is 0.05). Low p-values indicate the data did not occur by chance. For example, a p-value of 0.01 means there is only a 1% probability that the results from an experiment occurred by chance.

We have explained this in the paper.

  1. Also, I suggest to give fractions with "0" in the front. So instead of .05 it should be 0.05. It is much clearer.

We have changed the p-value to 0.05 instead of .05.

  1. Regarding section 5 ("Conclusion and Recommendations...") I think that the first paragraph should be moved to the previous section, as it does not convey any general thoughts, but details referring to the experiments. Also, I think that the last concluding paragraph should be rephrased to sound better in English, and to be sure that the authors wrote what they meant. I don't think the word "moreover" is appropriate when the authors jump from "the course too compact" to "potential in applications". 

We have reworded this content and have hired an English editing company to go through the rest of the manuscript.

Reviewer 2 Report

The core of the paper describes a result obtained from a practical experiment on 28 participants that aimed to explore the use of a visual programming language (in this case Scratch) to help developing computational thinking skills. The results seem to confirm this, but the paper does not provide sufficient detail in many important questions of study design, which undermines its utility.

Major issues:

MAJOR 1

The study considered no control group. How did the authors verify that any effect detected by the study did not come as a side effect of filling in the same questionnaire for the second time by the students?

MAJOR 2

I do not believe that such study (or any data in it) would be easily reproducible. The actual collected results are not published as a dataset (thus there is no way for anyone to review the assessment). The authors argue that this is for the "ethical reasons" but I find that argument invalid.  Given proper anonymization techniques in place, the data should pose no privacy danger at all; at the same time if the anonymization was not in place, the study (code evaluation and interview results) might have been flawed by any of the common personal biases.

At the same time, the methodology is described insufficiently. In this case I'd expect an appendix with precise questionnaires given to students (that is referred to around line 135 -- what are the modifications applied to the original design by Chien?). At the same time, the description of challenge tasks (in Table 1) is insufficient; although some of these are kind-of straightforward, all descriptions should be available precisely in the form given to the tested group. Some of the descriptions do not give any detail that could be used to at least recognize a working solution from a non-working one; e.g., Basic task 1 says to complete taskS (which ones?) using blocks with defined topics (which topics?).

Finally, if the developed learning platform is crucial for collecting the study results, it should be published along with the paper.

MAJOR 3

Statistical analysis might require reworking. Using a t-test on such a small sample might be problematic, but should be okay given the authors actually analyzed whether the preconditions for t-testing are satisfied (which I did not find). Reporting of the t-value is slightly uncommon; why not report the exact p-value directly to show the level of significance on these results?

Quite strikingly, I was not able to find any kind of assessment of the possible correlation between the improvement in student's coding skills and the actual outcome of the questionnaires.

MAJOR 4

While the English of writing is generally good, some parts of the article might require focusing on the main supportable findings of the research. Neither of introduction or conclusion provides a single unique view of any "main" hypothesis that would be confirmed or rejected by the study. In turn, while the study was successfully finished, the paper only serves as a record of this study being carried out and stays rather conclusion-less in the long-term.

Particularly, much of the introduction section (maker education etc.) is only partially related to the study outcome. Many arguments in the paper are also rather hand-wavy or under-specified; I was not able to find e.g. the meaning of "four core steps of CT" or the relation of the study results to "Inverted U-shape theory" (l.319).

Some minor issues:

Citation formatting could be improved (esp. last paragraph of Discussion).

Citations 1 and 21 are not archived by a publisher, self-archiving these as a supplementary material would be preferred.

Capitalization of some terms is sometimes confusing. (E.g., is the Visual Programming Language a concrete language or a generic term for visual programming languages?)

Author Response

The core of the paper describes a result obtained from a practical experiment on 28 participants that aimed to explore the use of a visual programming language (in this case Scratch) to help developing computational thinking skills. The results seem to confirm this, but the paper does not provide sufficient detail in many important questions of study design, which undermines its utility.

Major issues:

MAJOR 1

The study considered no control group. How did the authors verify that any effect detected by the study did not come as a side effect of filling in the same questionnaire for the second time by the students?

When both a control group and an experimental group are used, it is mainly to understand the differences between teaching activities. In this study we seek not to explore the difference between two teaching activities but instead to understand the difference in learning after the introducing the teaching experiment. Note that the following related studies use a single group and each still constitutes a valuable contribution to the literature.

  1. Daradoumis, T., Marquès Puig, J. M., Arguedas, M., & Calvet Liñan, L. (2022). Enhancing students’ beliefs regarding programming self-efficacy and intrinsic value of an online distributed Programming Environment. Journal of Computing in Higher Education, 1–31.
  2. Guven, G., Kozcu Cakir, N., Sulun, Y., Cetin, G., & Guven, E. (2022). Arduino-assisted robotics coding applications integrated into the 5E learning model in science teaching. Journal of Research on Technology in Education, 54(1), 108–126.
  3. Ke, F., Pachman, M., & Dai, Z. (2020). Investigating educational affordances of virtual reality for simulation-based teaching training with graduate teaching assistants. Journal of Computing in Higher Education, 32(3), 607–627.
  4. Cheng, K. H., & Tsai, C. C. (2019). A case study of immersive virtual field trips in an elementary classroom: Students’ learning experience and teacher-student interaction behaviors. Computers & Education, 140, 103600.
  5. Hu, C. C., Yeh, H. C., & Chen, N. S. (2020). Enhancing STEM competence by making electronic musical pencil for non-engineering students. Computers & Education, 150, 103840.
  6. Dubovi, I. (2022). Cognitive and emotional engagement while learning with VR: The perspective of multimodal methodology. Computers & Education, 183, 104495.

Although the students each completed the same questionnaire, the items were presented in a random order and the reliability and validity of the questionnaires was sufficiently high. During the experiment, we trust the answers provided by students and carry out an objective evaluation by using statistical analysis.

MAJOR 2

I do not believe that such study (or any data in it) would be easily reproducible. The actual collected results are not published as a dataset (thus there is no way for anyone to review the assessment). The authors argue that this is for the "ethical reasons" but I find that argument invalid.  Given proper anonymization techniques in place, the data should pose no privacy danger at all; at the same time if the anonymization was not in place, the study (code evaluation and interview results) might have been flawed by any of the common personal biases.

At the same time, the methodology is described insufficiently. In this case I'd expect an appendix with precise questionnaires given to students (that is referred to around line 135 -- what are the modifications applied to the original design by Chien?). At the same time, the description of challenge tasks (in Table 1) is insufficient; although some of these are kind-of straightforward, all descriptions should be available precisely in the form given to the tested group. Some of the descriptions do not give any detail that could be used to at least recognize a working solution from a non-working one; e.g., Basic task 1 says to complete taskS (which ones?) using blocks with defined topics (which topics?).

Finally, if the developed learning platform is crucial for collecting the study results, it should be published along with the paper.

First, most of the descriptions are provided in the paper. Second, our questionnaire is not anonymous. Third, if necessary, we can of course provide all of the data for reference. We have modified the Data Availability Statement.

In the appendix, we have provided all of the questionnaires. We have also described the content of each task in greater detail.

We have reorganized the section in this article that describes the learning system and added more content to clarify it.

MAJOR 3

Statistical analysis might require reworking. Using a t-test on such a small sample might be problematic, but should be okay given the authors actually analyzed whether the preconditions for t-testing are satisfied (which I did not find). Reporting of the t-value is slightly uncommon; why not report the exact p-value directly to show the level of significance on these results?
Quite strikingly, I was not able to find any kind of assessment of the possible correlation between the improvement in student's coding skills and the actual outcome of the questionnaires.

Thank you for the suggestion. We have provided the p values in the tables. Furthermore, this study mainly investigated the frequency of using CT steps rather than coding skills. Nevertheless, your suggestion is valuable for the future study.

MAJOR 4

While the English of writing is generally good, some parts of the article might require focusing on the main supportable findings of the research. Neither of introduction or conclusion provides a single unique view of any "main" hypothesis that would be confirmed or rejected by the study. In turn, while the study was successfully finished, the paper only serves as a record of this study being carried out and stays rather conclusion-less in the long-term.

Particularly, much of the introduction section (maker education etc.) is only partially related to the study outcome. Many arguments in the paper are also rather hand-wavy or under-specified; I was not able to find e.g. the meaning of "four core steps of CT" or the relation of the study results to "Inverted U-shape theory" (l.319).

Thank you for this suggestion. We have restructured and described the content, and have hired a professional English editing company to go through the article.

Some minor issues:

Citation formatting could be improved (esp. last paragraph of Discussion).

We have modified the citation format.

Citations 1 and 21 are not archived by a publisher, self-archiving these as a supplementary material would be preferred.

We have corrected these.

Capitalization of some terms is sometimes confusing. (E.g., is the Visual Programming Language a concrete language or a generic term for visual programming languages?)

We have corrected the capitalization.

Reviewer 3 Report

Hi Authors,

A detailed report is provided for your consideration.

1.       The paper should be revised with a section on the main contributions and the results achieved from the experimental studies. It is critical for the readers to understand the authors contributions.

2.       The research objectives are not clear to the reader.

3.       The paper lacks a framework that can highlight the novelty of the research. A VPL platform developed is useful to gain student attention. However, the process to analyse the data is based on standard industry practice.

4.       Figure 2 is not clear, replace with high resolution images. Applicable to all images.

5.       Line 198 mentions ‘The teacher could provide assistances or adjust the course content based on students’ capabilities’. How can this be achieved using the experimental study.

6.       The paper does not specify the criteria considered to select the participants. It is vital to mention the selection criteria and compare them with a baseline group.

7.       The paper does not provide insight on the factors that determine the critical thinking measurement of the participants. If it is manually reviewed the result can be biased and skewed. Is there an algorithm to evaluate the critical thinking score of the participants.

8.       The mental state of the participants should be considered to ensure the results were not impacted by their circumstances and state of mind. The age group considered in the test are vulnerable therefore integrating the study with attention span through Virtual Reality will assist in real time data capture.

9.       A brief description on t-test used to analyse the participants results will assist the readers understand the review process.  

10.   The teacher is required to review the back-end database to identify the issues faced by the participants. This process lacks novelty and can lead to biased decision making.   

11.   The paper lacks a literature review that can highlight the gaps in Critical thinking. There are multiple pedagogical studies that can studies and tests that can determine the behavioural pattern and critical thinking ability. Alternatively, the introduction of basic robotics programming to curriculum in the education industry helps improve critical thinking among school students.

12.   The tracking system which forms the basis of understanding the student requirements and developing customised training plans are not well explained.

13.   The teacher offers assistance to the students based on the learning status identified from the session and back-end data which is not novel.

14.   The paper repeatedly mentions four core steps of CT. However, the four steps are unclear, and it does not prove how the core steps assist in improving the problem solving skills of the participants.

Author Response

Hi Authors,

A detailed report is provided for your consideration.

  1. The paper should be revised with a section on the main contributions and the results achieved from the experimental studies. It is critical for the readers to understand the authors contributions.

According to the reviewer's suggestion, we have restructured and described the content, and have hired an English editing company to go through the manuscript.

  1. The research objectives are not clear to the reader.

We have restructured and reworded the content to clarify the article.

  1. The paper lacks a framework that can highlight the novelty of the research. A VPL platform developed is useful to gain student attention. However, the process to analyse the data is based on standard industry practice.

Thank you for the feedback. We have restructured and revised the article to emphasize the contribution of this paper.

  1. Figure 2 is not clear, replace with high resolution images. Applicable to all images.

We have corrected this figure.

  1. Line 198 mentions ‘The teacher could provide assistances or adjust the course content based on students’ capabilities’. How can this be achieved using the experimental study.

This was a mistake. We have fixed the paragraph.

  1. The paper does not specify the criteria considered to select the participants. It is vital to mention the selection criteria and compare them with a baseline group.

In our country, every third grader learns programming. This research focuses on a group of students in the upper grades of elementary school (fifth and sixth graders) who voluntarily participated in this course.

  1. The paper does not provide insight on the factors that determine the critical thinking measurement of the participants. If it is manually reviewed the result can be biased and skewed. Is there an algorithm to evaluate the critical thinking score of the participants.

This study does not explore critical thinking, nor does the term critical thinking appear in the article.

  1. The mental state of the participants should be considered to ensure the results were not impacted by their circumstances and state of mind. The age group considered in the test are vulnerable therefore integrating the study with attention span through Virtual Reality will assist in real time data capture.

Neither the learning environment nor the equipment used in this study could cause physical or mental harm to students. We consulted a number of experienced teachers concerning the content of the study. The content level is in line with the age and ability of the participants, and the participants could cease participating in the experiment at any time.

  1. A brief description on t-test used to analyse the participants results will assist the readers understand the review process.  

We have clarified this by restructuring and rewording this content.

  1. The teacher is required to review the back-end database to identify the issues faced by the participants. This process lacks novelty and can lead to biased decision making.   

Despite the many studies that proving the effectiveness of VPL for learning CT, most assess effectiveness using quantitative methods, either by CT tests or scales. Some assess students’ programming projects through systems. However, as such methods do not comprehensively analyze programming and learning processes, students’ operations during visual programming (VP) are not well understood. However, it is also important to effectively and automatically assess the learning effectiveness of CT and determine whether students exhibit an understanding of CT. In this study we tap the Scratch VPL as a programming language tool to teach CT, and develop a learning and tracking system for CT education. Using this system, programming projects and tasks can be mastered in real-time, and teachers can grasp the learning pace of every student as well as the results of the projects. Therefore, when the teacher observes the learning status, the system instantly analyzes, codes, and presents the results to the teacher.

We have restructured and reworded the content to clarify the article.

  1. The paper lacks a literature review that can highlight the gaps in Critical thinking. There are multiple pedagogical studies that can studies and tests that can determine the behavioural pattern and critical thinking ability. Alternatively, the introduction of basic robotics programming to curriculum in the education industry helps improve critical thinking among school students.

This study does not explore critical thinking, nor does the term critical thinking appear in the article. We have restructured and reworded the content to clarify the article.

  1. The tracking system which forms the basis of understanding the student requirements and developing customised training plans are not well explained.

We have added descriptions to clarify this part.

  1. The teacher offers assistance to the students based on the learning status identified from the session and back-end data which is not novel.

Despite the many studies that proving the effectiveness of VPL for learning CT, most assess effectiveness using quantitative methods, either by CT tests or scales. Some assess students’ programming projects through systems. However, as such methods do not comprehensively analyze programming and learning processes, students’ operations during visual programming (VP) are not well understood. However, it is also important to effectively and automatically assess the learning effectiveness of CT and determine whether students exhibit an understanding of CT. In this study we tap the Scratch VPL as a programming language tool to teach CT, and develop a learning and tracking system for CT education. Using this system, programming projects and tasks can be mastered in real-time, and teachers can grasp the learning pace of every student as well as the results of the projects. Therefore, when the teacher observes the learning status, the system instantly analyzes, codes, and presents the results to the teacher.

We have restructured and reworded the content to clarify the article.

  1. The paper repeatedly mentions four core steps of CT. However, the four steps are unclear, and it does not prove how the core steps assist in improving the problem solving skills of the participants.

We have added a description of the four core CT steps to clarify the process.

Reviewer 4 Report

This current paper is very interesting, it examined the learning effectiveness of applying the visual programming language integrated learning system to the computational thinking course. The  the idea is  attractive, and the design is relatively clear, but it needs further explanation. The most important thing is that the theoretical argumentation for the learning outcomes should be further introduced and further explored.

    Abstract: Line 12 to 26 should clearly introduce the research goal, the design and the theoretical questions relating to the target research goal. What’s more grammar mistakes in line 12 and 13 should be corrected. Line 15 and 16 is hard to understand, and should do some proofreading.

    Introduction: The paper should explain the theoretical rationale for the learning outcomes, for example the confidence motivation as the learning outcomes of the online competitional thinking courses. By the end of the introduction, the article should clearly introduce the research goal and research hypothesis in relation to the central target research question.

     Method: The label in figure one should use the verb plus noun. We should use the subject plus object to introduce the learning flow charts experiment procedure of figure one. And the article should insert the description of the task. The assessment tool, the detailed learning anxiety, learning confidence, should be introduced including reliability of Questionnaires. Questionnaire invented or modified in the article or as supplementary materials should be attached by the end of the paper. How the recorded and videotaped be transcribed should also be introduced.

     Research result:  3.1 FCTQ in page 208 should be introduced in full term, not abbreviated.  All the items from A1 to A9 should be clearly introduced and described. And then the reliability of the items should be clearly introduced in the survey. 

    Findings: The summary of the recent findings should be clearly introduced at the beginning of the discussion section. And then the theorical rationale for using all those learning analysis results, for example the motivation anxiety and the learning outcome should be clearly introduced again in relation to the effectiveness of the course. Further discussion on the analysis with analytical results should also be clearly introduced.

Author Response

This current paper is very interesting, it examined the learning effectiveness of applying the visual programming language integrated learning system to the computational thinking course. The idea is attractive, and the design is relatively clear, but it needs further explanation. The most important thing is that the theoretical argumentation for the learning outcomes should be further introduced and further explored.

Thank you for the comment.

    Abstract: Line 12 to 26 should clearly introduce the research goal, the design and the theoretical questions relating to the target research goal. What’s more grammar mistakes in line 12 and 13 should be corrected. Line 15 and 16 is hard to understand, and should do some proofreading.

We have restructured and reworded the content to clarify the article.

    Introduction: The paper should explain the theoretical rationale for the learning outcomes, for example the confidence motivation as the learning outcomes of the online competitional thinking courses. By the end of the introduction, the article should clearly introduce the research goal and research hypothesis in relation to the central target research question.

We have restructured and reworded the content to clarify the article, and have added the research questions in the Introduction section.

     Method: The label in figure one should use the verb plus noun. We should use the subject plus object to introduce the learning flow charts experiment procedure of figure one. And the article should insert the description of the task. The assessment tool, the detailed learning anxiety, learning confidence, should be introduced including reliability of Questionnaires. Questionnaire invented or modified in the article or as supplementary materials should be attached by the end of the paper. How the recorded and videotaped be transcribed should also be introduced.

According to the reviewer's suggestion, we have restructured and reworded the content and have hired a professional English editing company to go through the article. We also describe the tasks in detail and report the reliability value of the questionnaires. We have provided all the questionnaires in the appendix, and explained how the record is transcribed.

     Research result:  3.1 FCTQ in page 208 should be introduced in full term, not abbreviated.  All the items from A1 to A9 should be clearly introduced and described. And then the reliability of the items should be clearly introduced in the survey. 

FCTQ is Frequency of Using CT Skills Questionnaire. We describe the questionnaire in the Assessment Tools section and provide the questionnaires in the appendix.

    Findings: The summary of the recent findings should be clearly introduced at the beginning of the discussion section. And then the theorical rationale for using all those learning analysis results, for example the motivation anxiety and the learning outcome should be clearly introduced again in relation to the effectiveness of the course. Further discussion on the analysis with analytical results should also be clearly introduced.

According to the reviewer's suggestion, we have restructured and reworded the content to clarify the article.

Round 2

Reviewer 1 Report

I suggest to accept in the present (revised) version.

Author Response

Thanks!!

Reviewer 2 Report

After the revision, the paper seems generally much more enjoyable, with the main point being clear on the first view (and read). I still recommend a minor revision to clear out some of the last small issues, hoping that it would make the paper more useful, but other than that the study seems valid.

Minor points:

- please improve the captions of all figures and tables. The caption should ITSELF clearly explain the interpretation of the figure, and make sure that the figure can be understood even if it is the first thing that the reader sees from the paper. Some advice on that may be found in style guides of various journals. In case of tables, please include at least a short information on "how to interpret the data".

- I currently assume that the test methodology used to generate test values in the tables 3-6 is somehow verifiable or described in another research (the paper is still not very clear on what is compared), which should be stated/cited preferably right in the table caption. On the other hand, this can be partly inferred from the "research questions" stated in the first sections of the paper, so at least readers are able to arrive to some conclusion after a bit of search now.

- Some of the figures look like they suffered a bit of quality degradation from JPEG compression (esp. Fig 1). That might only be an issue of my copy of the manuscript draft, but please double check this doesn't appear in the final manuscript.

- Finally, I understand the "reasonable request" form of data availability statement and I find it quite reasonable to include as such, but we unfortunately live in a weird world where reason is a very flexible value. I saw many studies where by "reasonable request" authors meant anything ranging between "just send a mail" to "we might have a look at the mail and see later" and "don't even think we'll reply to you unless you send a business delegate with a hefty bag of money". I optimistically assume that the authors here meant something like "please send a mail that does not categorize as spam and doesn't put too much unnecessary load on the corresponding author", and that there might be some additional data disclosure limitations (e.g., NDA required, or data takedown policy signed). For the sake of reproducibility and science, please do clarify the data disclosure conditions precisely and in advance. I apologize for the nitpicking on this, sadly my current research area is plagued by this kind of data (un)availability.

Author Response

After the revision, the paper seems generally much more enjoyable, with the main point being clear on the first view (and read). I still recommend a minor revision to clear out some of the last small issues, hoping that it would make the paper more useful, but other than that the study seems valid.

A:Thank you for your suggestions to make our article better.

please improve the captions of all figures and tables. The caption should ITSELF clearly explain the interpretation of the figure, and make sure that the figure can be understood even if it is the first thing that the reader sees from the paper. Some advice on that may be found in style guides of various journals. In case of tables, please include at least a short information on "how to interpret the data".

A:We have modified the captions of all figures and tables and also included a short information to interpret the data.

I currently assume that the test methodology used to generate test values in the tables 3-6 is somehow verifiable or described in another research (the paper is still not very clear on what is compared), which should be stated/cited preferably right in the table caption. On the other hand, this can be partly inferred from the "research questions" stated in the first sections of the paper, so at least readers are able to arrive to some conclusion after a bit of search now.

A:First of all, we have mentioned the variables to be compared in the research question.

Second, in the experimental design part, this study also mentioned the variables to be analyzed.

Finally, the content of the analysis results in Tables 3~6 also explained the comparative variables and results. In addition, the appendix also provided the content of the questionnaire to let readers know more about the details of each questionnaire.

Some of the figures look like they suffered a bit of quality degradation from JPEG compression (esp. Fig 1). That might only be an issue of my copy of the manuscript draft, but please double check this doesn't appear in the final manuscript.

A:It may be a problem caused by the zooming of the picture, we will provide a picture with a clear resolution.

Finally, I understand the "reasonable request" form of data availability statement and I find it quite reasonable to include as such, but we unfortunately live in a weird world where reason is a very flexible value. I saw many studies where by "reasonable request" authors meant anything ranging between "just send a mail" to "we might have a look at the mail and see later" and "don't even think we'll reply to you unless you send a business delegate with a hefty bag of money". I optimistically assume that the authors here meant something like "please send a mail that does not categorize as spam and doesn't put too much unnecessary load on the corresponding author", and that there might be some additional data disclosure limitations (e.g., NDA required, or data takedown policy signed). For the sake of reproducibility and science, please do clarify the data disclosure conditions precisely and in advance. I apologize for the nitpicking on this, sadly my current research area is plagued by this kind of data (un)availability.

A :There are the following ways to write the Data Availability Statement in Sustainability, and we choose the most suitable content to write.

  1. Data Availability Statement: The data presented in this study are available on request from the corresponding author. The data are not publicly available due to privacy restrictions.
  2. Data Availability Statement: Not applicable.
  3. Data Availability Statement: The data presented in this study are available upon reasonable request from the corresponding author.
  4. Data Availability Statement: Most data is available online and cited in the reference list, except course handbooks from the UWTSD that we have not permission to share publicly.
  5. Data Availability Statement: The data that support the findings of this study are available from the corresponding author, upon reasonable request.

According to the reviewer comment, we have modified the content Data Availability Statement.

Reviewer 4 Report

The study is well done, but the language needs to be further revised. Some of the language problems are listed here,

in line 16-17, “we show that… “ should be “ the study shows”,

in line 20, the author says “we show…”, it should be “the study also shows that …” ,

in line 81, according to a survey by Zhang and Nouri can be deleted.

In line 82, the many studies, the should be deleted.

In line 81-82, “despite the many studies attesting the effectiveness of VPL for learning CT, most determine this using quantitative methods...” should be changed into “ despite the fact that many studies attests the effectiveness of …., most of them determine it by quantitative methods…”

In Line 84, through systems (what systems? Be specific.

In line 86, however, should be deleted.

In line 89, by which should be deleted.

In line 97, should be “ after participating in the course of …”.

In line 104, subjects should be changed into participants

In line 105 -106, should be “with the help of CT concepts, the participants completed tasks and challenges by using…”

In line 107, should be “the teacher had teaching experience in using Scratch…”

In line 122-123, the learners logged into the system to complete the tasks.

In line 185, during the last week of the experiment, should be changed into during the experiment of the previous week, we …

In line 263-265,  should be “in particular, when students observed their peers completing the tasks, they felt stressed, which manifested as learning anxiety.?

In line 27- , the increased should be deleted.

In line 275, analysis should be deleted.

In line 281-282,should be “ the first time when they attempted to solve the problems, they were…:

In line 283, should be problems, and tasks.

In line 290-291, the sentence” in addition to improving the learners’ …, CT concepts …” is hard to understand. Please make some changes according to its real meaning.

In line 301 to 302, should be success in solving…

In line 305-306. “As anxiety increases, performance increases,…and then becomes optimal when anxiety reaches a certain threshold,” what is the subject of “ and then becomes…” ? this sentence is hard to understand.

The whole text should be read carefully and language problems should be corrected carefully, before it can be published, 

Author Response

The study is well done, but the language needs to be further revised. Some of the language problems are listed here,

in line 16-17, “we show that… “ should be “ the study shows”,

in line 20, the author says “we show…”, it should be “the study also shows that …” ,

in line 81, according to a survey by Zhang and Nouri can be deleted.

In line 82, the many studies, the should be deleted.

In line 81-82, “despite the many studies attesting the effectiveness of VPL for learning CT, most determine this using quantitative methods...” should be changed into “ despite the fact that many studies attests the effectiveness of …., most of them determine it by quantitative methods…”

In Line 84, through systems (what systems? Be specific.

In line 86, however, should be deleted.

In line 89, by which should be deleted.

In line 97, should be “ after participating in the course of …”.

In line 104, subjects should be changed into participants

In line 105 -106, should be “with the help of CT concepts, the participants completed tasks and challenges by using…”

In line 107, should be “the teacher had teaching experience in using Scratch…”

In line 122-123, the learners logged into the system to complete the tasks.

In line 185, during the last week of the experiment, should be changed into during the experiment of the previous week, we …

In line 263-265,  should be “in particular, when students observed their peers completing the tasks, they felt stressed, which manifested as learning anxiety.?

In line 27- , the increased should be deleted.

In line 275, analysis should be deleted.

In line 281-282,should be “ the first time when they attempted to solve the problems, they were…:

In line 283, should be problems, and tasks.

In line 290-291, the sentence” in addition to improving the learners’ …, CT concepts …” is hard to understand. Please make some changes according to its real meaning.

In line 301 to 302, should be success in solving…

In line 305-306. “As anxiety increases, performance increases,…and then becomes optimal when anxiety reaches a certain threshold,” what is the subject of “ and then becomes…” ? this sentence is hard to understand.

The whole text should be read carefully and language problems should be corrected carefully, before it can be published, 

A: We have made changes as suggested by the reviewer.

Round 3

Reviewer 4 Report

the current paper is very interesting, it examined the learning effectiveness of applying the visual programming language integrated learning system to the computational thinking course. The paper is well written, the idea is very attractive , and  the authors have made certain modification according to the suggestions. However, the language still needs to be further modified before publication, for example, in line 103, the last word should be fifth.